# Instructor-inspired Machine Learning for Robust Molecular Property Prediction

**Fang Wu[1]\*, Shuting Jin[2], Siyuan Li[3], Stan Z. Li[3]**
[1] Computer Science Department, Stanford University
[2] School of Computer Science and Technology, Wuhan University of Science and Technology
[3] School of Engineering, Westlake University

## Abstract

Machine learning catalyzes a revolution in chemical and biological science. However, its efficacy heavily depends on the availability of labeled data, and annotating biochemical data is extremely laborious. To surmount this data sparsity challenge, we present an instructive learning algorithm named InstructMol to measure pseudo-labels' reliability and help the target model leverage large-scale unlabeled data. InstructMol does not require transferring knowledge between multiple domains, which avoids the potential gap between the pretraining and fine-tuning stages. We demonstrated the high accuracy of InstructMol on several real-world molecular datasets and out-of-distribution (OOD) benchmarks.

## 1   Introduction

An enduring obstacle in applied chemical and biological sciences is identifying and making chemical compounds or materials that have optimal properties for a given purpose [1, 2]. However, the vast majority of progress in these areas is still achieved primarily through time-consuming and costly trial-and-error experiments. Machine learning (ML) has undergone an unparalleled technological advancement, opening up a myriad of applications across various domains. Its potential is particularly notable in expediting the discovery and development of novel materials, pharmaceuticals, and chemical processes [3–5]. However, ML models' efficacy heavily relies on the availability of labeled data and the consistency of prediction targets. Meanwhile, the cost of generating new data labels through wet experiments is prohibitively high. Consequently, the size of labeled data in this field is several orders of magnitude lower than the one that can inspire breakthroughs in other ML fields [6]. This data scarcity severely hampers ML in addressing scientific challenges within this realm, impeding its ability to generalize to new molecules [7].

Biochemists have noticed this problem and propose several strategies to overcome the low data limitation (view Fig. 1). Firstly, inspired by the remarkable success of self-supervised learning from NLP [8] and CV [9, 10], researchers apply the pretrain and finetune paradigm [11–14] to molecule modeling. They boost the performance of various molecular models by pretraining them on massive unlabeled data. However, this benefit can be negligible when a large gap exists between the sample distributions of pretraining and downstream tasks [15]. An alternative option is to use active learning [16], which iteratively selects new data points to annotate according to the current model's predictions. However, auxiliary labor is still required to enrich the original database. Thirdly, domain knowledge is incorporated to enhance molecular representations, such as providing more high-quality hand-crafted features [17], constructing motif-based hierarchical molecular graphs [18], and leveraging knowledge graphs [19]. However, domain knowledge can be biased and is difficult to integrate into different training techniques universally.

---

\*Corresponding Author, email: `fangwu97@stanford.edu`

38th Conference on Neural Information Processing Systems (NeurIPS 2024).

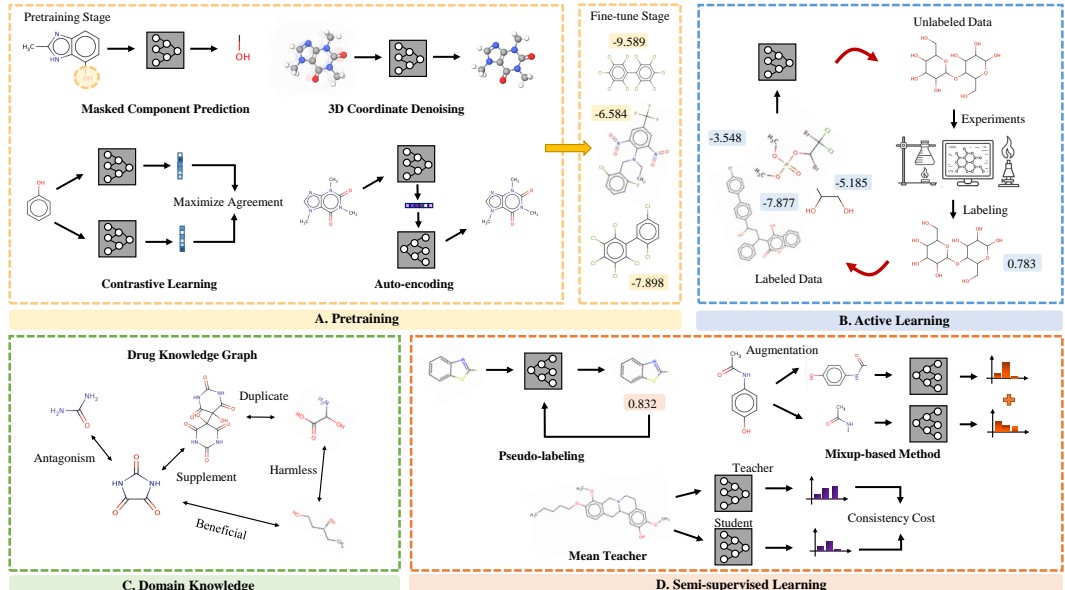

Figure 1: Four mainstream paradigms to ameliorate the scarcity of labeled biochemical data. (A) Self-supervised pretraining tasks include masked component modeling, contrastive learning, and auto-encoding. (B) Active learning involves the iterative selection of the most informative data, in which the molecular models are the most uncertain. These samples are then subjected to laboratory testing to determine their labels. This process is repeated with newly labeled data added to the training set. (C) Knowledge graphs are introduced to provide structured relations among multiple drugs and unstructured semantic relations associated with different drug molecules. (D) In SSL, the unlabeled data is used to create a smooth decision boundary between different classes or to estimate the distribution of the input data, while the labeled data is used to provide specific examples of the correct output.

In this study, we develop InstructMol, a flexible semi-supervised learning (SSL) approach to excavate the abundant unlabeled biochemical data for robust molecular property prediction. It differs from prior studies in two aspects: (1) it utilizes an additional instructor model to predict the confidence of the predicted label, measure its reliability, and generate pseudo-label information for unannotated data; (2) with the help of pseudo-label information guiding the model, the target molecular model can reliably utilize unlabeled data without the need to transfer knowledge between multiple domains, which perfectly avoids the potential gap between pre-training and fine-tuning stages. We demonstrate that InstructMol outpasses existing SSL approaches and achieves state-of-the-art performance in MoleculeNet and several OOD benchmark datasets. Besides, via InstructMol, we accurately predict the properties of all 9 newly discovered drug molecules in the latest patent (ZA202303678A). Extensive experiments showcase the effectiveness of our model in surmounting the challenge of data scarcity, propelling advancements in the chemistry and biology domains.

## 2   Related Work

How to exploit large-scale unlabeled molecular data becomes an essential topic in the ML community to alleviate the scarcity of labeled data and improve OOD generalization, where pretraining and SSL are the two major fast-growing lines. The former traditionally employ unsupervised techniques to pretrain ML models, such as autoencoder [20], autoregressive modeling [13, 21], masked component modeling [22, 23], context-prediction [24], contrastive learning [25], and multi-modality [26]. Despite several progress claims, the benefits of self-supervised pretraining can be negligible in many cases [15]. Recent years have witnessed a rising interest in developing SSL to reduce the amount of required labeled data [27]. Several hypotheses have been discussed in the literature to support specific SSL design decisions [28], such as the smoothness and manifold assumptions. Existing SSL algorithms can be roughly separated into three sorts: consistency regularization, proxy-label methods, and generative models. The consistency regularization is based on the simple concept that randomness

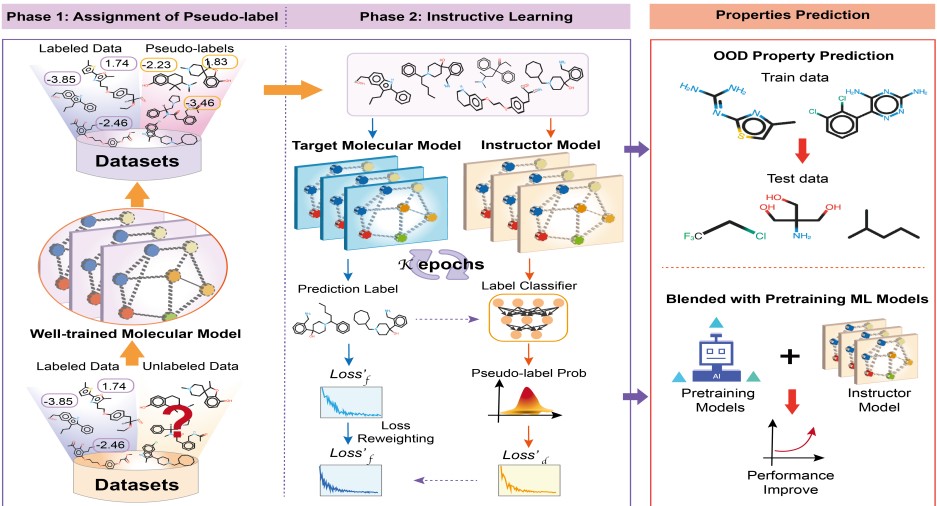

Figure 2: The outline of InstructMol. We utilize a pre-trained target molecular model to forecast the properties of unlabeled examples as pseudo-labels. Then, an instructor model predicts the confidence of those pseudo-annotations, which are leveraged to guide the target molecular model to distribute different attention in inferring different data points.

within the neural network (NNs) such as dropout) or data augmentation transformations should not modify model predictions given the same input and impose an auxiliary loss. This line of research includes $\pi$-model [29], temporal ensembling [29], and mean teachers [30] The proxy-label methods regard proxy labels on unlabeled data as targets and consist of two groups: self-training [31], where the model itself produces the proxy labels, and multi-view learning [32], where proxy labels are produced by models trained on different views of the data. The generative models rely on VAE [33] and GAN [34] to capture the joint distribution more accurately.

# 3 Preliminaries and Background

**Task formulation.** Suppose we have access to some labeled molecular data $\mathcal{D} = \{(x_i, y_i)\}_{i=1}^{N}$. $x_i$ can be in any kind of formats such as 1D sequences, 2D graphs, and 3D structures. $y_i$ can be any discrete (*e.g.*, toxicity, and drug reaction) or continuous properties (*e.g.*, water solubility, and free energy). Here we consider continuous property for simple illustration, but our approach can be easily extended to binary-class or multi-label circumstances. The target molecular model $f : \mathcal{X} \rightarrow \mathcal{Y}$ can be any category of architectures, including Transformers, GNNs, and geometric NNs. There are also some unseen data $\mathcal{D}^{test}$ to evaluate the performance of the learned model. Traditionally, $\mathcal{D}$ is divided into the train and validation sets as $\mathcal{D}^{train} = \left\{ \left( x_i^{train}, y_i^{train} \right) \right\}_{i=1}^{N_1}$ and $\mathcal{D}^{val} = \left\{ \left( x_i^{val}, y_i^{val} \right) \right\}_{i=1}^{N_2}$. Besides, we can obtain some unlabeled data points, denoted as $\mathcal{D}^{\star} = \{x_i^{\star}\}_{i=1}^{M}$, where the number of unlabeled data $M$ is usually orders of magnitude larger than that of labeled data $N$.

**Challenge of proxy-labeling.** Labeled data $\mathcal{D}$ and unlabeled data $\mathcal{D}^{\star}$ sometimes follow significantly different data distributions, *i.e.*, $\mathbb{P}(x_i, y_i) \neq \mathbb{P}(x_i^{\star}, y_i^{\star})$. However, many SSL algorithms [29, 35, 36] assume no distributional shift between $\mathcal{D}$ and $\mathcal{D}^{\star}$. They are likely only to reinforce the consistent information in the labeled data $\mathcal{D}$ to unlabeled examples $\mathcal{D}^{\star}$ instead of mining auxiliary information from $\mathcal{D}^{\star}$, without exception for proxy labeling [37]. More importantly, despite the versatility and modality-agnostic of proxy labeling, it achieves relatively poor performance compared to recent SSL approaches [38]. This arises because some pseudo-labels $\hat{y}_i^{\star}$ can be severely incorrect during training due to the poor generalization ability of classic DL models [39, 40]. If we directly utilize pseudo-labels from a previously learned model for subsequent training, the conformance-biased information in the proceeding epochs could increase confidence in erroneous predictions and eventually lead to a vicious circle of error accumulation [41, 42]. The situation can be even worse when labeled data $\mathcal{D}$ contain noises because of unavoidable experimental errors. Accordingly, it becomes essential to

grasp the quality and dependability of pseudo-annotations and intelligently select a subset of them to reduce the hidden noise.

**Motivation of InstructMol.** Though confidence is crucial for pseudo-label selection and this selection reduces pseudo-label error rates, NNs' poor calibration renders this solution insufficient. Explicitly, incorrect predictions in poorly calibrated NNs can also have high confidence (*i.e.*, $\hat{y}_i^\star \to 0$ or $\hat{y}_i^\star \to 1$) [38]. More importantly, prior studies such as UPS [38] resort to the target model's output $\hat{y}_i$ as the confidence indicator and produce hard labels by $y_i^\star = \mathbb{1}\left[\hat{y}_i^\star \geq \gamma_1\right]$ or $y_i^\star = \mathbb{1}\left[\hat{y}_i^\star \leq \gamma_2\right]$, where $\gamma_1$ and $\gamma_2$ are pre-defined thresholds. Nonetheless, this selection mechanism becomes inapplicable if $\mathcal{Y}$ is a continuous label space, as networks no longer output class probabilities. For regression tasks, $\hat{y}_i^\star$ discloses no confidential information, and numerous biochemical problems are regression-based, including molecular property prediction [43–45], 3D structure prediction [46], and binding affinity prediction [47]. This brings a challenge for probabilistic output-based proxy-labeling algorithms [38]. Thus, instead of depending on $\hat{y}_i^\star$ to judge proxy labels' reliability, we accompany the target molecular model $f$ with an additional instructor model $g$, which plays the role of a critic and predicts label observability, *i.e.*, whether the label is true or fake. $g$ disentangles the confidence prediction and the property prediction, reducing the noise introduced by the pseudo-labeling process.

## 4 Method

**The Overall Instructive Learning Framework.** We separate the integral workflow of InstructMol into two phases. In the first step, we retain pseudo-labels $\{\hat{y}_i^\star\}_{i=1}^M = f\left(\{\hat{x}_i^\star\}_{i=1}^M\right)$. There are several approaches to creating proxy labels, such as label propagation via neighborhood graphs [48]. Here, we require the molecular model $f$ to directly annotate samples in the unlabeled dataset $\mathcal{D}^\star$. Then, in the following step, we construct a new dataset with both labeled and pseudo-labeled samples as $\mathcal{D}' = \mathcal{D} \cup \{(x_i^\star, \hat{y}_i^\star)\}_{i=1}^M$ and proceed training both the target molecular model $f$ and the instructor model $g$ based on this new set. These two procedures are iteratively repeated until $f$ reaches the optimal performance on the validation set $\mathcal{D}^{val}$.

To be specific, the instructor model $g : (\mathcal{X} \times \mathcal{Y} \times \mathcal{H}_f) \to \mathcal{P}$ forecasts the confidence measure $p_i$ ($0 \leq p \leq 1$) of whether the given label $y_i'$ belongs to the ground-truth label set $\{y_i\}_{i=1}^N$ or the pseudo-label set $\{y_i^\star\}_{i=1}^M$. It digests three items: the data sample with its label $(x_i', y_i') \in \mathcal{D}'$ and an additional loss term $\mathcal{H}_f(f(x_i'), y_i')$, where $\mathcal{H}_f$ is traditionally selected as a root-mean-squared-error (RMSE) loss or a mean-absolute-error (MAE) loss for regression tasks and cross-entropy (CE) loss for classification problems. Here we regard $\mathcal{H}_f(.)$ as the ingredient of $g$'s input to provide more information about the main molecular property prediction task. At last, the instructor model $g$ is supervised via a binary CE loss (BCE) as:

$$\mathcal{L}_g\left(\mathcal{D}', \{\hat{y}_i'\}_{i=1}^{N+M}\right) = \sum_{(x_i', y_i') \in \mathcal{D}'} \mathrm{BCE}(p_i', c_i) = \sum_{(x_i', y_i') \in \mathcal{D}'} \mathrm{BCE}\left(g\left(x_i', y_i', \mathcal{H}_f\left(\hat{y}_i', y_i'\right)\right), c_i\right), \quad (1)$$

where $c_i \in [0, 1]$ is an integer and represents the observability mask. It indicates whether $y_i$ is pseudo-labeled ($c_i = 0$) or not ($c_i = 1$). However, since the number of unlabeled data $M$ is much larger than the number of labeled data $N$, it is proper to shift the loss function from BCE to a focal loss [49] (FL) for unbalanced classes as: $\mathcal{L}_g\left(\mathcal{D}', \{\hat{y}_i'\}_{i=1}^{N+M}\right) = \sum_{(x_i', y_i') \in \mathcal{D}'} \mathrm{FL}(p_i', c_i) = \sum_{(x_i', y_i') \in \mathcal{D}'} -(1 - p_i')^\gamma \log(p_i')$, where $\gamma \geq 0$ is a tunable focusing parameter.

Meanwhile, the target molecular model $f$ receives the discriminative information $\{p_i\}_{i=1}^{N+M}$ from the instructor model $g$ and uses it to reweight the importance of different samples in backpropagating its gradient. In other words, the instructor model $g$ guides the target model $f$ to deliver different attention to different labels so that correct labels are attached more importance while erroneous labels are ignored. This can be realized by specially designing the loss of the target model $f$ as:

$$\mathcal{L}_f\left(\mathcal{D}, \{p_i\}_{i=1}^{N+M}\right) = \alpha \sum_{(x_i, y_i) \in \mathcal{D}} \mathcal{H}_f(f(x_i), y_i) + \sum_{\left(x_j^\star, \hat{y}_j^\star\right) \in \mathcal{D}^\star} (2p_j - 1) \cdot \mathcal{H}_f\left(f\left(x_j^\star\right), \hat{y}_j^\star\right), \quad (2)$$

where $0 \leq \alpha \leq 1$ is a hyper-parameter to balance the dominance of labeled and unlabeled data sets. $\mathcal{L}_f(.)$ transforms the original main task into a cost-sensitive learning problem [50] by imposing

---
**Algorithm 1** InstructMol Algorithm
---
1: **Input:** target model $f$, instructor model $g$, labeled data $\mathcal{D}$, unlabeled data $\mathcal{D}^\star$, pseudo-label update frequency $k$, loss weight $\alpha$
2: Initialize and pretrain a target model $f_0$ and an instructor model $g_0$
3: **for** epochs $n = 0, 1, 2, ...$ **do**
4:    **if** $n \mod k == 0$ **then**
5:       With no gradient:
6:       $\hat{y}_i^\star \leftarrow f(x_i^\star), \quad \forall x_i^\star \in \mathcal{D}^\star$    ▷ Iteratively assign pseudo-labels every $k$ epochs
7:    **end if**
8:    $\mathcal{D}' \leftarrow \mathcal{D} \cup \{(x_i^\star, \hat{y}_i^\star)\}_{i=1}^M$    ▷ Construct the hybrid database
9:    $\hat{y}_i' \leftarrow f(x_i') \quad \forall x_i' \in \mathcal{D}'$
10:   $p_i \leftarrow g(x_i', y_i', \mathcal{H}_f(.)), \quad \forall (x_i', y_i') \in \mathcal{D}'$    ▷ Predict the confidence scores
11:   $\mathcal{L}_g\left(\mathcal{D}', \{\hat{y}_i'\}_{i=1}^{N+M}\right) \leftarrow$ Equation 1
12:   $\mathcal{L}_f\left(\mathcal{D}', \{p_i\}_{i=1}^{N+M}\right) \leftarrow$ Equation 2
13:   Update the parameters of $f$ and $g$ based on $\mathcal{L}_g(.)$ and $\mathcal{L}_f(.)$
14: **end for**
---

a group of soft-labeling weights based on the predicted confidence of data labels. That is, for pseudo-labeled samples (*i.e.*, $x_j' \in \mathcal{D}^\star$), the soft-labeling weight becomes $2p_j - 1$.

This loss format in Equation 2 induces different behaviors on the loss $\mathcal{H}_f(.)$ of labeled and pseudo-labeled instances, where the judgment $\{p_i\}_{i=1}^{N+M}$ produced by the instructor model $g$ is leveraged to differentiate their informativeness. Notably, if the instructor model $g$ regards a pseudo-label $\hat{y}_j^\star$ to be unreliable (*i.e.*, $0.5 > p_j > 0$), the loss $\mathcal{L}_f(.)$ chooses to enlarge the gap with the pseudo-label $\hat{y}_j^\star$. Meanwhile, once the instructor $g$ fully trusts the pseudo-label $\hat{y}_j^\star$, InstructMol forces the target molecular model $f$ to give more effort to inferring this proxy-labeled sample. Generally, the more likely a pseudo-label $\hat{y}_j^\star$ is considered reliable by the instructor model $g$ (*i.e.*, $p_j \rightarrow 1$), the stronger it drives the target molecular model $f$ to make further improvement on inferring this label $\hat{y}_j^\star$. Otherwise, InstructMol will push $f$ to overturn its previous belief if $p_j \rightarrow 0$. Noticeably, we can also impose a soft-labeling weight for samples with true labels (*i.e.*, $x_i' \in \mathcal{D}$). For instance, a weight factor of $\alpha \geq 1$ can navigate $f$ to concentrate more on samples that are more trusted by $g$. But we practically discover no significant refinement with this design on labeled data and leave it for future exploration. The whole pseudo-code of InstructMol is depicted in Algorithm 1.

**Loss Selection for InstructMol.** We compare some relevant methods from the literature under the proxy-labeling SSL algorithms in Appendix Tab. 6, containing the vanilla proxy-labeling (PL), curriculum learning for PL (CPL) [51], UPS, and self-interested coalitional learning (SCL) [52]. It is structured in three main columns that describe the selection for unlabeled samples, loss function, and fitness for regression problems. Except for SCL and InstructMol, all approaches adopt a subset of unannotated instances for training rather than utilizing the entire unlabeled datasets. SCL can take advantage of all data points from $\mathcal{D}^\star$, but its main limitation is its improper or even severely wrong loss function design. As the confidence score is close to 1, SCL assigns a strong negative multiplier factor as $1 - \frac{\alpha}{1-p_j} \rightarrow -\infty$ and forces the target model $f$ to disregard this label, which is actually reliable. On the contrary, when the instructor model $g$ doubts the reliability of $\hat{y}_j^\star$, the ratio $1 - \frac{\alpha}{1-p_j}$ becomes $1 - \alpha$, driving the target model $f$ to move towards it.

**Guidelines for InstructMol.** Before executing InstructMol, it is natural to first obtain a well-trained molecular target model $f_0$ through regular supervised learning on $\mathcal{D}$ and then initialize an instructor $g_0$ by discriminating pseudo-labels that are generated by $f_0$. This is empirically proven to achieve higher training stability and robustness. Moreover, pseudo-labels are assigned every $k$ epoch and a proper setting of $k$ is critical to the success of InstructMol. If pseudo-labels are updated too frequently, the training procedure will be volatile at the very beginning. While a too-large $k$ would significantly increase the training complexity. Here we adopt an adaptive decay strategy: with an initial value $k_0$, the update frequency decreases by a factor of 0.5 until it reaches the minimum threshold $k_{\min} = 3$.

**Analysis of InstructMol.** After the curation of $\mathcal{D}'$, there are two distinct learning tasks during the second stage of InstructMol. Specifically, the target model follows the regular routine to predict the molecular properties. Meanwhile, the instructor model strives to differentiate whether the label

Table 1: Performance of three distinct ML models with different SSL methods on nine molecular property prediction tasks. For classification tasks, we calculate the ROC-AUC, while for regression tasks, we use RMSE as the evaluation metric. The number in the bracket is the standard deviation of three runs.

| | Classification (ROC-AUC %, higher is better ↑) | | | | | | Regression (RMSE, lower is better ↓) | | |
|---|---|---|---|---|---|---|---|---|---|
| Datasets
# Molecules
# Tasks | BBBP
2039
1 | BACE
1513
1 | ClinTox
1478
2 | Tox21
7831
12 | ToxCast
8575
617 | SIDER
1427
27 | ESOL
1128
1 | FreeSolv
642
1 | Lipo
4200
1 |
| **GIN** | 65.6(0.2) | 76.2(0.5) | 76.1(0.5) | 74.2(0.4) | 61.1(0.1) | 59.0(0.8) | 1.955(0.023) | 0.897(0.010) | 0.740(0.018) |
| + Semi-GAN [34] | 66.1(0.3) | 76.8(0.8) | 76.4(0.7) | 74.5(0.6) | 62.0(0.2) | 59.7(1.1) | 1.907(0.038) | 0.884(0.011) | 0.725(0.026) |
| + $\pi$-model [29] | 66.3(0.3) | 76.7(0.7) | 76.5(0.7) | 74.7(0.6) | 62.3(0.2) | 59.8(1.0) | 1.895(0.040) | 0.881(0.013) | 0.710(0.024) |
| + UPS [38] | 67.0(0.4) | 78.1(0.8) | 77.3(0.6) | 75.2(0.7) | 66.1(0.8) | 62.4(1.6) | – | – | – |
| **+ InstructMol** | **70.5(0.5)** | **83.3(0.8)** | **86.2(0.6)** | **76.6(0.3)** | **67.0(1.1)** | **64.2(1.2)** | **1.771(0.015)** | **0.839(0.018)** | **0.652(0.040)** |
| **GAT** | 64.8(0.1) | 77.9(0.3) | 69.3(0.3) | 72.2(0.4) | 61.7(0.1) | 55.9(0.6) | 2.069(0.011) | 0.866(0.009) | 0.813(0.022) |
| + Semi-GAN [34] | 64.9(0.2) | 78.1(0.5) | 69.6(0.4) | 72.3(0.4) | 61.9(0.3) | 56.2(0.7) | 2.017(0.034) | 0.852(0.017) | 0.802(0.025) |
| + $\pi$-model [29] | 65.3(0.2) | 78.5(0.4) | 69.7(0.6) | 72.6(0.3) | 62.2(0.4) | 56.5(0.8) | 1.959(0.046) | 0.847(0.021) | 0.780(0.029) |
| + UPS [38] | 67.2(0.6) | 80.4(0.5) | 74.9(1.2) | 74.1(1.0) | 65.7(1.5) | 59.3(1.4) | – | – | – |
| **+ InstructMol** | **68.1(0.4)** | **82.5(1.1)** | **77.3(1.0)** | **74.8(0.8)** | **66.4(1.7)** | **60.8(1.5)** | **1.862(0.017)** | **0.825(0.013)** | **0.738(0.028)** |
| **GCN** | 62.4(0.1) | 73.8(0.4) | 76.3(0.2) | 73.6(0.1) | 64.5(0.7) | 61.2(0.6) | 2.245(0.014) | 0.842(0.011) | 0.756(0.015) |
| + Semi-GAN [34] | 62.6(0.2) | 74.0(0.6) | 76.6(0.3) | 74.0(0.2) | 64.8(0.9) | 61.5(0.9) | 2.198(0.015) | 0.835(0.019) | 0.744(0.018) |
| + $\pi$-model [29] | 62.7(0.2) | 74.4(0.5) | 76.6(0.4) | 74.3(0.1) | 65.0(1.0) | 61.4(0.8) | 2.146(0.020) | 0.833(0.023) | 0.737(0.022) |
| + UPS [38] | 65.8(0.4) | 79.0(0.8) | 82.2(0.8) | 75.1(1.1) | 66.5(1.9) | 63.2(2.1) | – | – | – |
| **+ InstructMol** | **67.6(0.3)** | **80.1(1.0)** | **84.0(0.6)** | **75.7(0.9)** | **67.3(1.5)** | **64.7(1.9)** | **1.975(0.018)** | **0.811(0.020)** | **0.701(0.017)** |

Table 2: In-domain and OOD performance on the GOOD benchmark All numerical results are averages across 3 random runs.

| | GOOD-HIV ↑ | | | | GOOD-PCBA ↑ | | | |
|---|---|---|---|---|---|---|---|---|
| | scaffold | | size | | scaffold | | size | |
| | ID | OOD | ID | OOD | ID | OOD | ID | OOD |
| ERM | 82.79 | 69.58 | 83.72 | 59.94 | 33.45 | 16.89 | 34.31 | 17.86 |
| IRM [57] | 81.35 | 67.97 | 81.33 | 59.00 | 33.56 | 16.90 | 34.28 | 18.05 |
| VREx [58] | 82.11 | 70.77 | 83.47 | 58.53 | 33.88 | 16.98 | 34.09 | 17.79 |
| GroupDRO [59] | 82.60 | 70.64 | 83.79 | 58.98 | 33.81 | 16.98 | 33.95 | 17.59 |
| DANN [60] | 81.18 | 70.63 | 83.90 | 58.68 | 33.63 | 16.90 | 34.17 | 17.86 |
| Deep Coral [61] | 82.53 | 68.61 | 84.70 | 60.11 | 33.47 | 16.93 | 34.49 | 17.94 |
| Q-SAVI [62] | 82.73 | 70.66 | 84.58 | 59.92 | 33.90 | 16.88 | 34.76 | 17.99 |
| Mixup [63] | 82.29 | 68.88 | 83.16 | 59.03 | 30.22 | 16.59 | 30.63 | 17.06 |
| DIR [64] | 82.54 | 67.47 | 80.46 | 57.11 | 32.55 | 14.98 | 32.89 | 16.61 |
| **InstructMol** | **84.16** | **72.10** | **85.44** | **63.97** | **36.02** | **18.55** | **35.91** | **19.20** |

is real. Notably, some prior works embody a similar idea of jointly learning multiple tasks. For instance, the multi-objective optimization (MOO) methods [53] exploit the shared information and the underlying commonalities between two tasks and solve the problem by minimizing an augmented loss. In addition, GAN [54] makes a generator and a discriminator constantly compete against each other. Nevertheless, MOO cannot handle possible contradictions among different tasks in certain settings, where jointly minimizing the augmented loss may impede both tasks from attaining the global optimal [55]. While GAN has been praised for generating high-quality data, it is notoriously difficult to train and requires a large amount of training data. More essentially, making predictions in an adversarial manner for unlabeled data deviates from our primary goal of pseudo-labeling but is a mature technology for domain adaptation [56].

# 5 Experiments

We carry out a wide scope of experiments in all contexts. Section 5.1 shows the benefits of InstructMol in predicting molecular properties compared with various SSL learning algorithms. Section 5.2 verifies the superiority of InstructMol in lowering the predictive error over existing OOD generalization algorithms. Section 5.3 investigates the possibility of marring InstructMol with cutting-edge pretraining methods and demonstrates its potency by setting a state-of-the-art performance on MoleculeNet. Section 5.4 analyzes pseudo-labels' accuracy, the instructor's behavior, and ablation studies.

## 5.1 Molecular Property Prediciton

**Data and Setups.** We first investigate the effectiveness of InstructMol on three sorts of backbone including GCN [65], GAT [66], and GIN [67], and report their performance on the standard MoleculeNet [43]. Datasets are divided using scaffold splitting into training, validation, and test sets with a

Table 3: Performance on molecular property prediction tasks, where **GEM+InstructMol** achieves the best result.

| | Classification (ROC-AUC %, higher is better ↑) | | | | | | Regression (RMSE, lower is better ↓) | | |
|---|---|---|---|---|---|---|---|---|---|
| Datasets | BBBP | BACE | ClinTox | Tox21 | ToxCast | SIDER | ESOL | FreeSolv | Lipo |
| # Molecules | 2039 | 1513 | 1478 | 7831 | 8575 | 1427 | 1128 | 642 | 4200 |
| # Tasks | 1 | 1 | 2 | 12 | 617 | 27 | 1 | 1 | 1 |
| **w.o. pretraining** | | | | | | | | | |
| D-MPNN [68] | 71.0(0.3) | 80.9(0.6) | 90.6(0.6) | 75.9(0.7) | 65.5(0.3) | 57.0(0.7) | 1.050(0.008) | 2.082(0.082) | 0.683(0.016) |
| Attentive FP [69] | 64.3(1.8) | 78.4(0.02) | 84.7(0.3) | 76.1(0.5) | 63.7(0.2) | 60.6(3.2) | 0.877(0.029) | 2.073(0.183) | 0.721(0.001) |
| MGCN [70] | 65.0(0.5) | 73.4(0.8) | 90.5(0.4) | 74.1(0.6) | – | 58.7(1.9) | – | – | – |
| **w. pretraining** | | | | | | | | | |
| N-Gram$_{RF}$ [71] | 69.7(0.6) | 77.9(1.5) | 77.5(4.0) | 74.3(0.4) | – | 66.8(0.7) | 1.074(0.107) | 2.688(0.085) | 0.812(0.028) |
| N-Gram$_{XGB}$ [71] | 69.1(0.8) | 79.1(1.3) | 87.5(2.7) | 75.8(0.9) | – | 65.5(0.7) | 1.083(0.082) | 5.061(0.744) | 2.072(0.030) |
| PretrainGNN [72] | 68.7(1.3) | 84.5(0.7) | 72.6(1.5) | 78.1(0.6) | 65.7(0.6) | 62.7(0.8) | 1.100(0.006) | 2.764(0.002) | 0.739(0.003) |
| InfoGraph [73] | 69.2(0.8) | 73.9(2.5) | 75.1(5.0) | 73.0(0.7) | 62.0(0.3) | 59.2(0.2) | – | – | – |
| GPT-GNN [22] | 64.5(1.1) | 77.6(0.5) | 57.8(3.1) | 75.3(0.5) | 62.2(0.1) | 57.5(4.2) | – | – | – |
| GROVER$_{base}$ [74] | 70.0(0.1) | 82.6(0.7) | 81.2(3.0) | 74.3(0.1) | 65.4(0.4) | 64.8(0.6) | 0.983(0.090) | 2.176(0.052) | 0.817(0.008) |
| GROVER$_{large}$ [74] | 69.5(0.1) | 81.0(1.4) | 76.2(3.7) | 73.5(0.1) | 65.3(0.5) | 65.4(0.1) | 0.895(0.017) | 2.272(0.051) | 0.823(0.010) |
| 3D-Infomax [75] | 69.1(1.1) | 79.4(1.9) | 59.4(3.2) | 74.5(0.7) | 64.41(0.9) | 53.37(3.4) | 0.894(0.028) | 2.337(0.227) | 0.739(0.009) |
| GraphMVP [26] | 72.4(1.6) | 81.2(0.9) | 79.1(2.8) | 75.9(0.5) | 63.1(0.4) | 63.9(1.2) | 1.029(0.033) | – | 0.681(0.010) |
| MolCLR [25] | 72.2(2.1) | 82.4(0.9) | 91.2(3.5) | 75.0(0.2) | – | 58.9(1.4) | 1.271(0.040) | 2.594(0.249) | 0.691(0.004) |
| Uni-Mol [76] | 72.9(0.6) | 85.7(0.2) | 91.9(1.8) | 79.6(0.5) | 69.6(0.1) | 65.9(1.3) | 0.788(0.029) | 1.620(0.035) | 0.603(0.010) |
| GEM [24] | 72.4(0.4) | 85.6(1.1) | 90.1(1.3) | 78.1(0.1) | 69.2(0.4) | 67.2(0.4) | 0.798(0.029) | 1.877(0.094) | 0.660(0.008) |
| **GEM + InstructMol** | **73.3(0.8)** | **85.9(1.3)** | **92.5(2.1)** | **79.9(0.6)** | **70.8(0.4)** | **67.4(0.9)** | **0.761(0.043)** | **1.604(0.043)** | **0.582(0.010)** |

ratio of 8:1:1. We report the mean and standard deviation of the results for three random seeds. More experimental details are put in Appendix A.2.

**Baselines.** Several SSL baselines are chosen including consistency regularization, proxy-label methods, and generative models. $\pi$-**model** [29] applies different stochastic transformations (*i.e.*, dropout) to the networks instead of the input graphs. **Semi-GAN** [34] introduces a discriminator to classify whether the input is labeled or not. Uncertainty-aware pseudo-label selection (**UPS**) [38] leverages the prediction uncertainty to guide the pseudo-label selection procedure but is merely applicable to classification problems.

**Results.** Tab. 1 shows that InstructMol significantly improves various ML architectures and outperforms all SSL baselines. For GIN, GAT, and GCN, it leads to an average increase in ROC-AUC of 8.6%, 7.1%, and 6.6%, respectively, for six classification tasks and an average decrease in RMSE of 9.3%, 8.0%, and 7.7% separately for three regression tasks. These statistics demonstrate our approach effectively boosts existing ML models in low-data circumstances for molecular scaffold property prediction, as most datasets in MoleculeNet have only thousands of labeled samples. Besides that, more up-to-date ML models like GIN enjoy stronger benefits of our InstructMol than primitive ones like GCN. It is also worth mentioning that InstructMol overcomes UPS's drawbacks and can be utilized for regression tasks. All evidence clarifies that InstructMol is a more advanced proxy labeling algorithm than the existing SSL mechanisms with stronger virtual screening capacity and broader applications.

## 5.2 OOD Generalization

**Data and Setsups.** Measuring OOD generalization is particularly relevant in molecular property prediction, where distributional shifts can be large and difficult to handle for ML models. Different molecular datasets obtained by distinct pharmaceutical companies and research groups often contain compounds from vastly different areas of chemical space that exhibit high structural heterogeneity. Towards this goal, we leverage the Graph Out-of-Distributio (GOOD) benchmark [77], where **GOOD-HIV** is a small-scale dataset for HIV replication inhibition and **GOOD-PCBA** includes 128 bioassays and forms 128 binary classification tasks. They are divided into a training set, an in-domain (ID) validation set, an ID test set, and OOD test sets by covariate and concept shift splits.

**Baselines.** The empirical risk minimization (**ERM**) and several OOD algorithms are considered. **IRM** [57] searches for data representations that perform well across all environments by penalizing feature distributions with different optimal linear classifiers for each environment. **VREx** [58] targets both covariate robustness and the invariant prediction. **GroupDRO** [59] tackles the problem that the distribution minority lacks sufficient training. **DANN** [60] adversarially trains the regular classifier and a domain classifier to make features indistinguishable. **Deep Coral** [61] encourages features in different domains to be similar by minimizing the deviation of covariant matrices from different

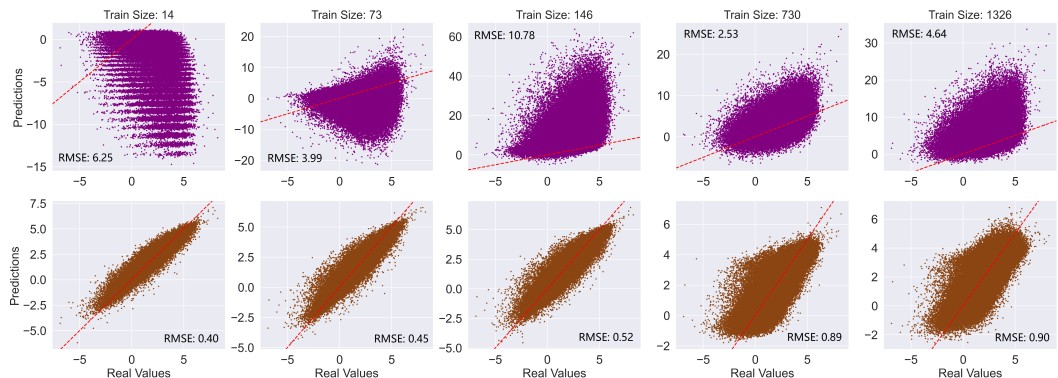

Figure 3: The scatter plot of the distributions of LogP predictions for unlabeled data with and without InstructMol. The first row includes predictions before instructive learning, and the second row includes predictions after instructive learning.

domains. **Q-SAVI** [62] encodes explicit prior knowledge of the data-generating process into a prior distribution over functions. **DIR** [64] and **Mixup** [63] are two graph-specific OOD methods.

**Results.** Tab. 2 documents the statistics, where most OOD algorithms have comparable performance with ERM. The risk interpolation methods like GroupDRO and the risk extrapolation mechanisms like VREx perform favorably against others on multiple shift splits. In contrast, InstructMol significantly exceeds ERM and other OOD baselines in all circumstances. Specifically, InstructMol brings improvements of 2.40%, 2.05%, 7.68%, and 4.69% for GOOD-HIV and GOOD-PCBA's different ID splits, and 3.62%, 6.72%, 9.83%, and 7.51% for GOOD-HIV and GOOD-PCBA's different OOD splits. It can be seen that gains for OOD are greater than benefits for ID, indicating the promise of pseudo-labeling to tackle OOD generalization.

### 5.3 SSL with Self-supervised Learning

**Setups and Background.** Pretraining and SSL are not mutually exclusive but can collaborate for more robust scientific investigations [78]. So we design a two-step workflow: (1) In the pretraining stages, unlabeled data is first used in a task-agnostic way, and we attain more general molecular representations. Then, those general representations are adapted for a specific task for fine-tuning. (2) In the instructive learning stage, unlabeled data is used again in a task-specific way via InstructMol. We combine GEM [24] and InstructMol and examine their joint effectiveness on MoleculeNet.

**Baselines.** Multiple baselines are selected for a thorough comparison. D-MPNN [68], MGCN [70] and AttentiveFP [69] are supervised GNN methods. N-gram [71], PretrainGNN [72], InfoGraph [73], GPT-GNN [22], GROVER [74], 3D-Infomax [75], GraphMVP [26], MolCLR [25], and Uni-Mol [76] are pretraining methods. We adopt the same scaffold splitting strategy as GEM and Uni-Mol with three repeated runs.

**Results.** The overall performance of InstructMol based on GEM and other baseline methods is summarized in Tab. 3. InstructMol achieves new SOTA results on all MoleculeNet tasks and brings an average improvement of 4.35%. Notably, its benefits are much stronger for regression tasks with a mean decrease of 9.98% in RMSE. Another set of results on MoleculeNet with a different splitting method is in Appendix Tab. 5, where InstructMol also outperforms all baselines. This highlights the necessity and importance of leveraging unlabeled examples to refine and transfer task-specific knowledge after pretraining through instructive learning. It also implies that InstructMol is not incompatible with existing pre-training ML models but can effectively supplement them with an additional instructor model.

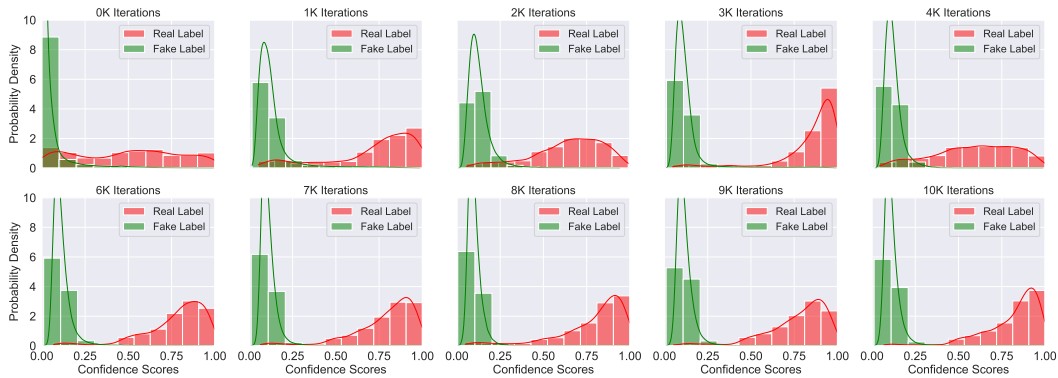

Figure 5: The distributions of confidence scores given by the instructor model during the training process.

## 5.4 Discussion, Ablation, and Other Applications

**Empirical evidence of pseudo-labels' accuracy.** Pseudo-labels' accuracy is crucial, as it reflects InstructMol's ability to generalize to unseen molecules. However, no ground-truth annotations exist for unlabeled data across the mentioned tasks. To address this issue, we adopt partition-coefficient values (LogP) as the target. A key predictor of drug-likeness featured in the famous "Lipinski's rule of five," LogP can quickly screen large libraries of potential therapeutics using computational tools like XLogP. We utilize a public Kaggle dataset of 14.6k molecules with associated LogP values for training and evaluate the predictions on the much larger ZINC15, whose LogP is computed using RDKit. Furthermore, measurements often number in the tens rather than thousands in the low-data regime of drug discovery. To assess InstructMol's limits, we significantly reduce the training size and investigate its efficacy with only 0.1%, 0.5%, 1%, 5%, and 10% of the entire training samples, resulting in scaffold-split training sets of 14, 73, 146, 730, and 1,326 molecules, respectively.

Fig. 3 compares the LogP prediction distributions before and after instructive learning. It can be observed that DL models trained solely on labeled data perform poorly in estimating unseen molecules' LogP with an average RMSE of 5.63. Contrarily, InstructMol enables a pretty accurate prediction of LogP with an average RMSE of 0.63 even with a very limited number of training examples, verifying the robustness of pseudo-labels.

**Ablation Studies.** We further investigate the effects of the number of unannotated molecules on the performance of InstructMol. As shown in Figure 4, the enrichment of unlabeled data consistently brings benefits to various downstream tasks.

**Real-world Drug Discovery.** We retrieved the latest patent ZA202303678A targeting the 5-HT1A receptor to examine IntructMol in real-world applications. ZA202303678A is published after 2023, and the property test standards of new compounds in ZA202303678A are consistent with those recorded in the CHEMBL214_Ki dataset [79]. All 9 new small molecule drugs are marked as good binders by InstructMol. Since the patent ZA202303678A provides accurate ground truth values of $K_i$ through wet experiments, we compare the predicted values with the real ones as shown in Appendix 7. It can

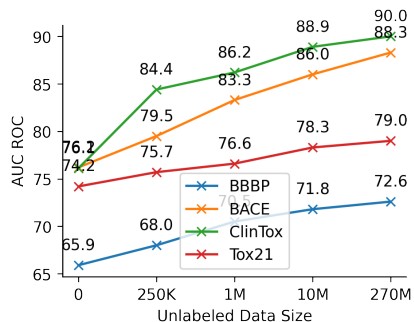

Figure 4: The influence of unlabeled data size on four tasks.

be observed that the errors are, at most, within four times, and most predicted results are close to real values. Among them, predicted $K_i$ of (S)-5-FPT (Ground truth: $K_i = 4$, Prediction: $K_i = 3.71$), (S)-5-NaT (Ground truth: $K_i = 64$, Prediction: $K_i = 61.86$), (S)-5-CPPT (Ground truth: $K_i = 0.6$, Prediction: $K_i = 0.93$), and (S)-5-FPyT (Ground truth: $K_i = 1.3$, Prediction: $K_i = 1.29$) is almost equal to actual $K_i$ obtained from biological experiments.

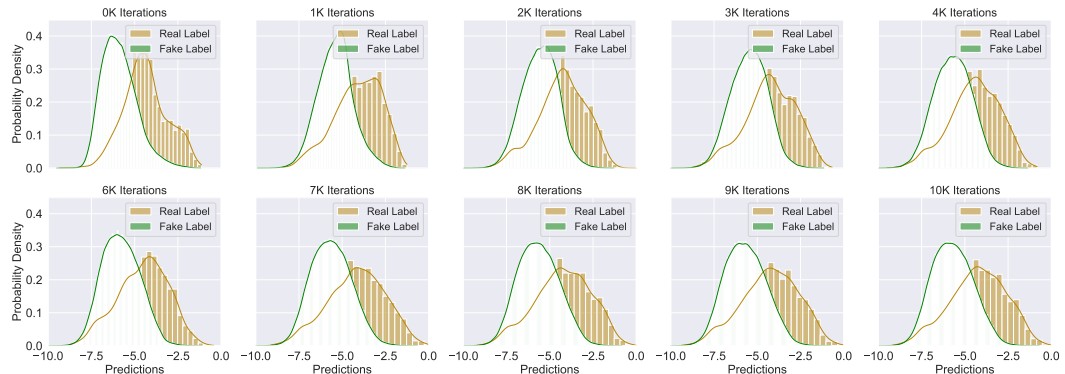

Figure 6: The distribution of the predictions of labeled data and unlabeled data generated by the target model during different training stages.

**Instructor model's Behavior.** The instructor estimates the target model's uncertainty. Therefore, thoroughly evaluating its judgment and contribution is beneficial. Here, we analyze its output value distribution on labeled and unlabeled data in different training stages of an activity cliff estimation task (CHEMBL214_Ki). The plots in Fig. 5 demonstrate the transition of an instructor model. Notably, since we pretrain the instructor before SSL, it performs well in distinguishing fake labels but remains confused with real ones. From the first iterations to 10K iterations, the instructor gradually gains a stronger capacity to discriminate true labels (confidence score $\rightarrow$ 1.0) and fake ones (confidence score $\rightarrow$ 0.0). Taking a step further, we draw the distribution of the target model's output in Fig. 6 and show a significant overlap between predictions of labeled and unlabeled data. This undoubtedly excludes the hypothesis that the instructor discriminates labeled and unlabeled data simply based on distinct distributions of their predictions. Even though predictions of labeled and unlabeled data are highly similar, the instructor still succeeds in comprehending their uncertainty and guides the target model to leverage pseudo-labels more cautiously. The instructor model's interpretability is a byproduct of our InstructMol and can be useful for many real-world biochemical problems. Here we give the relevant appendix figures cited in the manuscript.

# 6 Conlcusion

This paper presents InstructMol, a novel instructive learning framework, to alleviate the difficulty of experimentally obtaining the ground truth properties of molecular data and to overcome the limitation of the small number of labeled biochemical data points. InstructMol sufficiently cutting-edge pretraining methods for molecular representation, and addresses essential real-world problems.

**Limitations.** Despite the great progress of InstructMol in achieving enhanced molecular learning capacity, there are still minor limitations that require future exploration. For instance, our combination of InstructMol with self-supervised mechanisms is based on existing methodologies such as GEM, Uni-Mol, and GROVE. It is interesting to develop a more suitable self-supervised learning algorithm that can be better aligned with InstructMol.

**Acknowledgments.** This work was supported by the National Natural Science Foundation of China (No.62402351), the Hubei Provincial Natural Science Foundation of China (No.2024AFB275), and the Scientific Research Project of Education Department of Hubei Province (No.Q20231109).

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

# Appendices

## Appendix A  Experimental Setups

### A.1  Unlabeled Data

We use the ZINC15 [80] database to collect unlabeled molecular data, which can be downloaded from DeepChem [81]. There are four different data sizes supported by ZINC15: 250K, 1M, 10M, and 270M. For MoleculeNet, we utilize the 1M molecules as the unlabeled dataset. Those unlabeled SMILES are then converted by RDKit [82] into 2D graphs. We expect future work to enrich the unlabeled corpus by leveraging other resources such as ChemBL [83], Chembridge, and Chemdiv.

### A.2  Molecular Property Predictions

**Dataset Description.**   We conduct experiments on the MoleculeNet [43] to examine the efficaciousness of our algorithm for molecular property prediction. It is a widely used benchmark and we include 9 datasets in the main text, which are described as follows:

- **BBBP**. The blood-brain barrier penetration (BBBP) dataset contains binary labels of blood-brain barrier penetration (permeability).
- **BACE**. The BACE dataset provides quantitative ($IC_{50}$) and qualitative (binary label) binding results for a set of inhibitors of human $\beta$-secretase 1 (BACE-1).
- **ClinTox**. The ClinTox dataset compares drugs approved by the FDA and those that have failed clinical trials for toxicity reasons.
- **Tox21**. The "Toxicology in the 21st Century" (Tox21) initiative created a public database, which contains qualitative toxicity measurements on 12 biological targets, including nuclear receptors and stress response pathways.
- **Toxcast**. ToxCast is another data collection (from the same initiative as Tox21) providing toxicology data for a large library of compounds based on in vitro high-throughput screening, including experiments on over 600 tasks.
- **SIDER**. The Side Effect Resource (SIDER) is a database of marketed drugs and adverse drug reactions (ADR), grouped into 27 system organ classes.
- **ESOL**. ESOL is a small dataset consisting of water solubility data (log solubility in mols per liter) for common organic small molecules.
- **FreeSolv**. The Free Solvation Database (FreeSolv) provides experimental and calculated hydration-free energy of small molecules in water.
- **Lipo**. Lipophilicity is an important feature of drug molecules, which affects both membrane permeability and solubility. This dataset provides experimental results of octanol/water distribution coefficient The Free Solvation Database (FreeSolv) provides(logD at pH 7.4).

**Data Split.**   In our experiment, we follow the previous work GEM [24] and Uni-Mol [76] and adopt the scaffold splitting to divide different datasets into training, validation, and test sets with a ratio of 80%, 10%, and 10%. It has been widely acknowledged that scaffold splitting is more challenging than random splitting because the scaffold sets of molecules in different subsets do not intersect [76]. This splitting tests the model's generalization ability and reflects the realistic cases [43], and [76] find that whether or not chirality is considered when generating the scaffold using RDKit has a significant impact on the division results. The performance of different methods in Table 1 all follows the same scaffold splitting, where MolCLR is reproduced. In all experiments, we choose the checkpoint with the best validation loss and document the results on the test set run by that checkpoint.

**Implementation Details.**   In our experiments for molecular property prediction, we utilize 4 A100 GPUs and an Adam Optimizer [84] with a weight decay of 1e-16 for all GNN models, *i.e.*, GCN, GAT, and GIN. A ReduceLROnPlateau scheduler is employed to automatically adjust the learning rate with a patience of 10 epochs. Before the SSL stage, we first pretrain the target molecular model via supervised learning for 100 epochs and then pretrain the instructor model for 50 epochs, where an

early stopping mechanism is utilized with a patience of 5 epochs. Similar to GEM [24], we normalize the property label by subtracting the mean and dividing the standard deviation of the training set. As for the performance of baselines, we copy all available results from GEM [24], Uni-Mol [76] and 3D-Infomax [75].

As for the reproduction of three SSL methods, Semi-GAN is modified from `https://github.com/opetrova/SemiSupervisedPytorchGAN`. $\pi$-model is transformed from a simple Tensorflow-based version at `https://github.com/geosada/PI`. UPS is directly modified from its official GitHub at `https://github.com/nayeemrizve/ups`.

**Hyperparameter Search Space.** Referring to prior studies, we adopt a grid search to find the best combination of hyperparameters for the molecular property prediction task. To reduce the time cost, we set a smaller search space for the large datasets (*e.g.*, ToxCast). We report the details of the hyperparameter setup of InstructMol in Table 4.

Table 4: Hyperparameters setup for InstructMol in molecular property prediction.

| Hyperparameters Search Space | Symbol | Value |
|---|---|---|
| **Training Setup** | | |
| Epochs | – | [100, 200, 300] |
| Batch size | – | [32, 64, 128] |
| Learning rate | – | [1e-4, 5e-5, 1e-6, 5e-6] |
| Warmup ratio | – | [0.0, 0.05, 0.1] |
| The initial update frequency | $k_0$ | [5, 10, 20] |
| Unlabeled data size | – | [1K, 10K, 100K, 250K] |
| Balance weight for Uulabeled data and labeled data | $\alpha$ | [0.01, 0.05, 0.1, 0.2, 0.3, 0.5] |
| **GNN Architecture** | | |
| Dropout rate | – | [0.2, 0.4] |
| Number of GNN layers | – | [2, 3, 4, 5, 6] |
| The hidden dimension of node representations | – | [32, 64, 128, 256, 512] |
| The hidden dimension of edge representations | – | [64, 128, 256] |
| Number of heads in GMT | – | [4, 8 ,12] |
| Hidden dimension in GMT | – | [64, 128, 256, 512] |
| Number of fully-connected layers | – | [1, 2] |

### A.3 LogP Value Prediction

The Kaggle dataset for LogP prediction is downloaded from `https://www.kaggle.com/datasets/matthewmasters/chemical-structure-and-logp?resource=download`.

## Appendix B  Additional Experiments

### B.1 Performance for Random Scaffold Splitting

We additionally execute experiments using the random scaffold splitting on the classification datasets following the same experimental setting used in GROVE [74], which is much easier than the standard scaffold splitting used in our main text. As shown in Table 5, our findings notice that GEM blended with InstructMol also achieves stronger results than all baselines. The baseline results are copied from GROVE and GEM.

## Appendix C  Comparison among Proxy-labeling Algorithms

Here we provide a clear and simple comparison between InstructMol and existing proxy-labeling approaches in Table 6.

Table 5: Comparison of performance on the molecular property prediction tasks, where a random scaffold splitting is adopted.

| Datasets | BBBP | BACE | ClinTox | Tox21 | ToxCast | SIDER | Avg. |
|---|---|---|---|---|---|---|---|
| **Classification** (ROC-AUC %, higher is better ↑ ) | | | | | | | |
| # Molecules | 2039 | 1513 | 1478 | 7831 | 8575 | 1427 | – |
| # Tasks | 1 | 1 | 2 | 12 | 617 | 27 | – |
| **w.o. pretraining** | | | | | | | |
| D-MPNN | 91.9(3.0) | 85.2(5.3) | 89.7(4.0) | 82.6(2.3) | 71.8(1.1) | 63.2(2.3) | 80.7 |
| Attentive FP | 90.8(5.0) | 86.3(1.5) | 93.3(2.0) | 80.7(2.0) | 57.9(0.1) | 60.5(6.0) | 78.3 |
| **w. pretraining** | | | | | | | |
| N-Gram$_{XGB}$ | 91.2(1.3) | 87.6(3.5) | 85.5(3.7) | 76.9(2.7) | – | 63.2(0.5) | – |
| PretrainGNN | 91.5(4.0) | 85.1(2.7) | 76.2(5.8) | 81.1(1.5) | 71.4(1.9) | 61.4(0.6) | 77.8 |
| GROVER$_{base}$ | 93.6(0.8) | 87.8(1.6) | 92.5(1.3) | 81.1(1.5) | 72.3(1.0) | 65.6(0.6) | 82.3 |
| GROVER$_{large}$ | 94.0(1.9) | 89.7(2.8) | 94.4(2.1) | 81.9(2.0) | 72.3(1.0) | 65.8(2.3) | 83.4 |
| GEM | 95.3(0.7) | 92.5(1.0) | 97.7(1.9) | 83.1(2.5) | 73.7(1.0) | 66.3(1.4) | 85.2 |
| GEM + InstructMol | **95.8(1.4)** | **93.2(1.6)** | **97.8(2.0)** | **83.7(3.0)** | **74.2(1.3)** | **67.0(1.8)** | **85.3** |

Table 6: Comparison over different literature under the proxy-labeling framework.

| Method | Unlabeled Sample Selection | Loss Function | Reg. |
|---|---|---|---|
| Vanilla PL | $\mathbb{1}\left[\hat{y}_i^\star \geq \gamma_1\right] + \mathbb{1}\left[\hat{y}_i^\star \leq \gamma_2\right]$ | $\mathcal{H}_f\left(f\left(x_j^\star\right), \hat{y}_j^\star\right)$ | No |
| CPL | $\mathbb{1}\left[\hat{y}_i^\star \geq \gamma_1(e)\right] + \mathbb{1}\left[\hat{y}_i^\star \leq \gamma_2(e)\right]$ | $\mathcal{H}_f\left(f\left(x_j^\star\right), \hat{y}_j^\star\right)$ | No |
| UPS | $\mathbb{1}\left[u(\hat{y}_i^\star) \leq \mu_1\right]\mathbb{1}\left[\hat{y}_i^\star \geq \gamma_1\right] + \mathbb{1}\left[u(\hat{y}_i^\star) \leq \mu_2\right]\mathbb{1}\left[\hat{y}_i^\star \leq \gamma_2\right]$ | $\mathcal{H}_f\left(f\left(x_j^\star\right), \hat{y}_j^\star\right)$ | No |
| InstructMol | All Samples from $\mathcal{D}^\star$ | $(2p_j - 1) \cdot \mathcal{H}_f\left(f\left(x_j^\star\right), \hat{y}_j^\star\right)$ | Yes |

## C.1 Discovery of Drug-like Molecules

It should be noted that these 9 new small molecules are highly similar in structures, but IntructMol successfully and confidently differentiates their bioactivity based on their geometries. In addition to that, most experienced medicinal chemists also believe that the two sets of data within 10 times are manipulable and can be recognized as accurate prediction results. The error range of InstructMol's results for the prediction of small molecule properties of new compounds further proves the potential of our model in predicting essential properties of activity cliffs molecular and can effectively get aware of subtle changes in the presence of bioactivity cliffs.

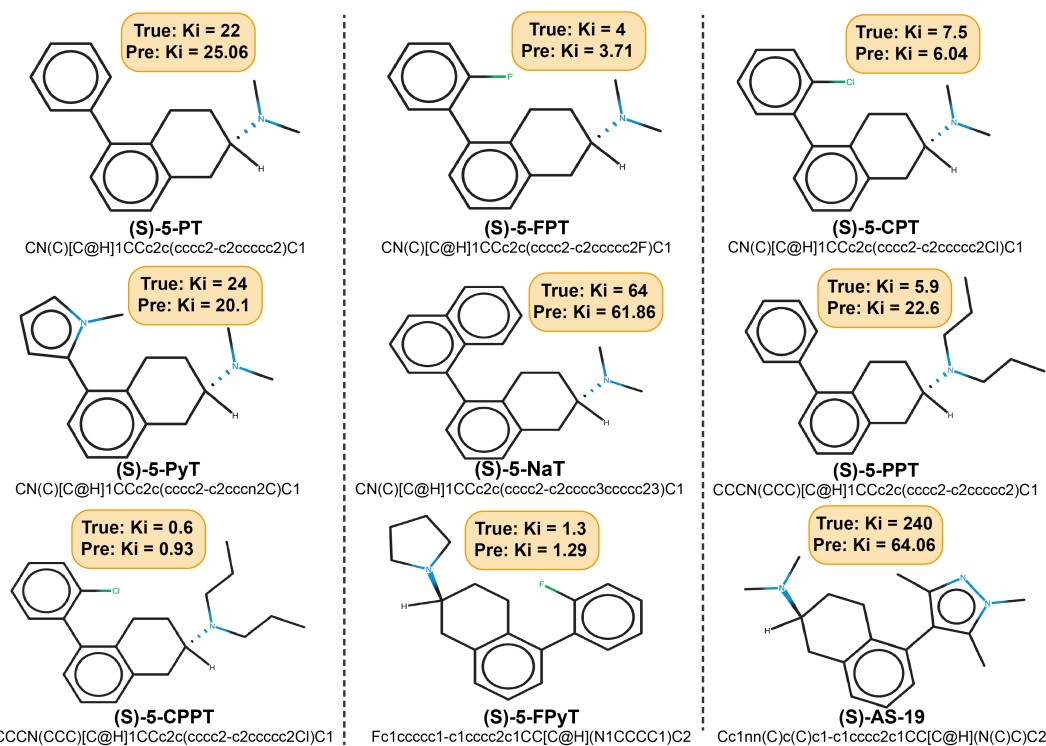

Figure 7: The $K_i$ value prediction results of the 9 newly discovered small molecules of 5-HT1A receptor.

