# OpenReview forum: "Instructor-inspired Machine Learning for Robust Molecular Property Prediction"
_NeurIPS.cc/2024/Conference — NeurIPS 2024 poster_

### Official Review · Reviewer_UbNR · 2024-06-12

**Soundness:** 4
**Presentation:** 4
**Contribution:** 3
**Rating:** 7
**Confidence:** 4

**Summary:**

This paper proposes a framework InstructMol for utilizing unlabeled data to help molecular property prediction on out-of-distribution (OOD) domains. The framework combines (1) a molecular model *f* that predicts (pseudo)labels with (2) a binary classifier *g* as instructor that evaluates the probability of labels being pseudo and reweighs *f*’s loss. In the experiments, it is compared to several self-supervised learning (SSL) baselines in predictive accuracy and OOD generalization. The experiments also investigate the effects of unlabeled data size and the instructor model’s behavior as an uncertainty estimator.

**Strengths:**

- Utilizing large, unlabeled, and potentially distribution-shifted data is relevant in chemical and materials sciences, and this paper presents an effective method for that.
- This paper is well-written, the figures are informative, and the experiments are comprehensive.

**Weaknesses:**

- Sec. 2’s review on related works is comprehensive but could be better organized.
  - The separation of pretraining and SSL is confusing: techniques like contrastive learning are often viewed as SSL, and some SSL algorithms work as pretraining.
  - The mention of methods “jointly learning multiple tasks” (Line 171) seems more appropriate in related works than method.
- The paper presents strong empirical results but not enough interpretations. Compared to SSL, what extra information does InstructMol utilize, or why does it extract information more effectively, that leads to better accuracy/generalizability? Discussing these could provide more insights to future model development.

**Questions:**

- In Line 33, domain knowledge being biased is raised as a problem. (1) Could some “bias” be helpful, e.g., by providing correct inductive bias to the model? (2) I suggest citing relevant literature on the scenarios where bias becomes a problem.
- Line 78, “usually follow significantly different distributions” seems exaggerating. Is there justification?
- Line 97, could uncertainty quantification methods, e.g., Bayesian NN, SNGP, evidential DL, address the problem for regression tasks?
- Section titles of 5.3 and 5.4 (Visualization) are confusing.
- The meaning of “confidence score” is unclear: in the main text (Lines 156, 301), 1 indicates pseudo-label, while Fig. 5 seems the opposite.

**Limitations:**

Discussed in Appendix.

---

> ### Author Rebuttal · Authors · 2024-07-31
>
> Dear Reviewer **UbNR**,
>
> Thank you for your comprehensive and insightful review of our paper. We appreciate your recognition of the strengths and contributions of our work, as well as your constructive feedback on areas for improvement. We are pleased that you found our framework effective in utilizing large, unlabeled, and potentially distribution-shifted data, which is indeed crucial in the fields of chemical and materials sciences. Your positive feedback on the clarity of our writing, the informativeness of the figures, and the comprehensiveness of our experiments is greatly encouraging.
>
> (1) We acknowledge that the review in Section 2 could be better organized. We will restructure this section to more clearly differentiate between different categories of related works, including a more distinct separation of pretraining and SSL. We will also ensure that our discussions on techniques like **contrastive learning** are appropriately classified and explained within the broader context of SSL. Besides, we will move the discussion on **jointly learning multiple tasks** in Line 171 to the related work section according to your suggestion.
>
> (2) We understand the need for a deeper discussion on why InstructMol performs better compared to existing SSL methods, particularly in terms of the extra information it utilizes or extracts. From our humble point of view, InstructMol introduces a **cooperative-yet-competitive learning** scheme to jointly boost the performance of both the main molecular prediction task and the companion label confidence prediction task. Specifically, InstructMol forges collaboration between both tasks by providing extra information to each other, *i.e.*, the main task provides prediction loss while the companion task provides label confidence measures. This is considered the major factor for the performance improvements.
>
> (3) Regarding the point on bias, we agree that some correct inductive biases can indeed provide beneficial inductive biases to the model. For instance, [A, B] construct chemical element knowledge graphs to summarize microscopic associations between elements and facilitate contrastive learning efficiency. Nevertheless, bias can lead to issues sometimes. [C] mentions that the regular distance threshold (*e.g.*,4-5 Angstrom ) to determine the chemical bond between atoms and build the graph connectivity can result in suboptimal performance. We will elaborate on this aspect, providing citations to relevant literature and clarifying the context in which we consider bias problematic.
>
> **[A]** Knowledge graph-enhanced molecular contrastive learning with functional prompt. Nature Machine Intelligence 2023.
>
> **[B]** Molecular contrastive learning with chemical element knowledge graph. AAAI 2022.
>
> **[C]** Discovering and Explaining the Representation Bottleneck of Graph Neural Networks from Multi-order Interactions. IEEE TKDE 2024.
>
> (4) The sensitivity of SSL approaches to distribution shifts between the labeled and unlabeled data has long been a crucial topic. Several prior studies [A,B,C] have demonstrated that SSL models suffer when the distribution of the unlabeled data differs significantly from the labeled data. [D] notice that the performance of SSL methods can degrade under distribution shifts and propose a method (FixMatch) to mitigate this issue by using strong data augmentation techniques. However, we value your comment and will tone down the language used (*usually* to *sometimes*) and provide a more nuanced discussion, supported by examples or references, to justify the statement about different distributions in molecular data.
>
> **[A]** Realistic Evaluation of Deep Semi-Supervised Learning Algorithms. NeurIPS 2018.
>
> **[B]** Using Self-Supervised Learning Can Improve Model Robustness and Uncertainty. NeurIPS 2019.
>
> **[C]** Unsupervised Representation Learning by Predicting Image Rotations. ICLR 2018.
>
> **[D]** FixMatch: Simplifying Semi-Supervised Learning with Consistency and Confidence. NeurIPS, 2020.
>
>
> (5) Your suggestion to explore uncertainty quantification methods like Bayesian Neural Networks (BNNs), SNGP, and evidential DL is absolutely valuable. For instance, BNNs [A] provide a probabilistic approach to neural networks by estimating the distribution over the network's weights rather than relying on a single set of weights. This probabilistic framework allows BNNs to quantify uncertainty in predictions, which is crucial for estimating the confidence of a regression model. Despite the great potential of BNNs in quantifying the uncertainty, they usually show inferior performance compared to DL methods. We look forward to future efforts in exploring BNNs and other promising mechanisms to take advantage of the abundant unlabeled molecular database.
>
> **[A]** Uncertainty quantification of molecular property prediction with Bayesian neural networks. arXiv 2019.
>
> (6) we have revised the titles of Sections 5.3 and 5.4 (**Discussion, Ablation, and Other Applications**) to more accurately reflect their content and avoid confusion. Additionally, we state that 0 and 1 indicate pseudo-labels and ground truths, respectively, and acknowledge a typo in Line 301. The correct version would be *...to discriminate true labels (confidence score $\rightarrow$ 1.0) and fake ones (confidence score $\rightarrow$ 0.0)*. As for Line 156, we have double-checked our analysis and corrected it to "...forces the target model $f$ to disregard this label, which is actually reliable". Thanks for your advice on terminology and section titles!

---

> > ### Comment · Reviewer_UbNR · 2024-08-09
> >
> > I appreciate the authors' comprehensive response, which addresses my concerns well. I am keeping my scores for this paper high at this stage.

---

> > > ### Author Response · Authors · 2024-08-13
> > > **Thanks for Feedback**
> > >
> > > Dear Reviewer UbNR,
> > >
> > > We are delighted that you have found our rebuttal feedback valuable and maintained your positive score. We greatly appreciate your recognition of our work and would respectfully inquire if there might be any additional opportunities for us to improve our manuscript. Once again, thanks for your constructive feedback, and we would eagerly welcome any further guidance at your convenience!
> > >
> > > Best regards,
> > >
> > > Authors

---

### Official Review · Reviewer_pmns · 2024-07-09

**Soundness:** 3
**Presentation:** 3
**Contribution:** 3
**Rating:** 6
**Confidence:** 4

**Summary:**

The paper introduces InstructMol, an innovative learning framework designed to address the challenge of data sparsity in chemical and biological sciences by leveraging large-scale unlabeled data through reliable pseudo-labeling. Unlike traditional methods that rely on transferring knowledge between domains, InstructMol operates within a single domain, eliminating potential discrepancies between pretraining and fine-tuning stages. The authors demonstrate the algorithm's high accuracy using various real-world molecular datasets and out-of-distribution (OOD) benchmarks, demonstrating its effectiveness in enhancing machine learning applications in biochemical research.

**Strengths:**

- Well-written and easy to follow
- InstructMol effectively addresses the data scarcity issue in biochemical data by leveraging large-scale unlabeled data without the need for domain transfer.
- Extensive experiments are conducted to demonstrate the efficacy of InstructMol.

**Weaknesses:**

- The paper does not provide a detailed analysis of the computational complexity or resource requirements of the InstructMol algorithm.
- While the paper showcases the superior performance of InstructMol, it does not sufficiently address potential overfitting issues that may arise due to the iterative use of pseudo-labels.
- Although the paper compares InstructMol with several baseline methods, it does not include a thorough comparison with some of the latest advancements in semi-supervised learning and domain adaptation techniques.

**Questions:**

- Due to limited discussion on computational complexity, it difficult to assess the practicality and scalability of the approach for very large datasets or in resource-constrained environments.
- It was mentioned that determining 'k' in InstructMol is important; please show the experimental results and discussion related to this.
- It would be beneficial to have an experiment to determine whether using a poor model for model $f$ can still result in performance compensation due to the confidence score, or if it leads to a decline in performance.
- Why RMSEs in Figure 3 become worse as the training data increases?
- What model and data were used for training on "Real-word Drug Discovery"?

**Limitations:**

The authors addressed limitations in Appendix.

---

> ### Author Rebuttal · Authors · 2024-08-01
>
> Dear Reviewer **pmns**,
>
> Thank you for your detailed review and insightful feedback on our paper, "InstructMol." We appreciate your recognition of the strengths, particularly in addressing data scarcity in biochemical research and the clarity of our presentation. We are glad that you found the paper well-written and that our approach effectively addresses data sparsity by leveraging large-scale unlabeled data. Your positive remarks on the comprehensiveness of our experiments are encouraging.
>
> (1) We recognize the importance of addressing the practical scalability of InstructMol. The computational cost of the algorithm primarily involves the instructor model and the iterative pseudo-labeling process. Since pseudo-labeling is applied every $k$ epoch, the theoretical complexity of InstructMol is approximately $(1 + 1/k)$ times that of a standard semi-supervised learning (SSL) approach, which trains only the target model. However, in practical implementation, we have observed that InstructMol converges significantly faster than existing SSL methods. For example, in the classification tasks reported in Table 1, the UPS [A] method typically requires around 100 epochs for convergence, whereas InstructMol completes training in approximately 10-20 epochs. This accelerated convergence offers a clearer perspective on the practical implications of using InstructMol on large-scale datasets or in environments with limited computational resources.
>
> **[A]** In Defense of Pseudo-Labeling: An Uncertainty-Aware Pseudo-label Selection Framework for Semi-Supervised Learning. ICLR 2021.
>
> (2) The parameter $k$ plays a crucial role in the InstructMol framework. We agree with your point that it is necessary to present the experimental results that explore the impact of different values of 'k' on the model's performance. The table below shows the influence of different $k$ strategies, where **na** indicates a loss explosion without convergence and the number in the bracket corresponds to the number of epochs for convergence. It can be found that a too frequent update would make the training procedure volatile, resulting in training failure, while a large $k$ would significantly increase the training complexity. In contrast, our adaptive decay strategy (ADS) achieves a competitive performance while maintaining a fast training speed.
>
> | k | BBBP  | BACE | ClinTox |
> |---|---|---| ---|
> | 1  | na (--) | 77.9 (9)  | na |
> | 10  | 70.7 (37) | 83.1 (25)  | 86.5 (42)|
> | ADS | 70.5 (13) | 83.3 (11) | 86.2 (18) |
>
> (3) The concern about using a suboptimal model and its impact on performance is valid. This is also one of the major reasons that we conduct experiments in Table 1. To be specific, all backbone architectures, containing GIN [A]/GCN [B]/GAT [C], are simple and "old-school" graph-based neural architectures, which were invented many years ago. Their performance is also far worse than state-of-the-art baselines such as Graphormer [D] and GPS [E]. However, our experimental results demonstrate that even when the base model is not optimal,  the confidence scores generated by the instructor model can also positively affect the final performance. This analysis helps in understanding the robustness of InstructMol to variations in model quality.
>
> **[A]** How Powerful are Graph Neural Networks? ICLR 2018.
>
> **[B]** Graph Attention Networks. NeurIPS 2018.
>
> **[C]** Semi-Supervised Classification with Graph Convolutional Networks. ICLR 2017.
>
> **[D]** Do Transformers Really Perform Bad for Graph Representation? NeurIPS 2021.
>
> **[E]** Recipe for a general, powerful, scalable graph transformer. NeurIPS 2022.
>
>
> (4) The observation that RMSEs worsen as training data increases may suggest issues such as noise in the additional data or potential overfitting. Importantly, adding more training data does not automatically enhance a model's generalization to the test set. If the distribution of the additional training samples significantly differs from that of the test domain, this can lead to an out-of-distribution (OOD) transfer problem, resulting in poorer performance on the test set. However, Figure 3 demonstrates that our InstructMol framework can mitigate the distributional gap introduced by new data, thereby enhancing the model's robustness.
>
>
> (5) Thanks for your question about the real-world drug discovery application. We employ the CHEMBL214_Ki dataset from the ACE benchmark [A] and GEM [B] as the base architecture, which shows extraordinary results in molecular property prediction.
>
> **[A]** Exposing the limitations of molecular machine learning with activity cliffs. JCIM 2022.
>
> **[B]** Geometry-enhanced molecular representation learning for property prediction. Nature 2022.

---

> > ### Comment · Reviewer_pmns · 2024-08-12
> >
> > I appreciate the authors' time and effort. While they have addressed most of my concerns, some aspects remain unclear.
> >
> > - In the experiments related to the proposed \( k \) in the rebuttal, there are still questions about the effectiveness of the adaptive decay strategy.
> > - I agree that adding training data can introduce more noise. However, I still have concerns that this may indicate a potential weakness in the model's robustness to noise.

---

### Official Review · Reviewer_nboj · 2024-07-12

**Soundness:** 3
**Presentation:** 3
**Contribution:** 3
**Rating:** 6
**Confidence:** 4

**Summary:**

This paper targets the problem of label-scarcity in the domain of molecular property prediction. It can be seen as an improved version of proxy labeling. It utilizes a separate model that measures pseudo-labels’ reliability and helps the target model leverage large-scale unlabeled data. This method applies to both classification and regression tasks. The authors run numerous experiments on predicting molecular properties, OOD generalization, and combination with pre-training models.

**Strengths:**

The target problem is both timely and important, with a well-justified motivation. The proposed method alleviates some issues by utilizing an instructor model to predict confidence scores.

Overall, the method is clearly presented, and adequate experiments are conducted and documented.

**Weaknesses:**

The improvement in model performance is relatively weak, with large standard deviations, and the method is compute-demanding due to the separate model and the iterative procedure. Additionally, some technical details and experimental results are unclear.

**Questions:**

* In Line 201-202, how are the average increase in AUC-ROC and the average decrease in MSE defined and calculated? Why are there only three numbers for the six classification tasks?
* I assume the architecture of the InstructMol model is not necessarily the same as the target molecular model. The article does not clearly explain how the InstructMol model is constructed.

**Limitations:**

The authors briefly touched upon the necessity of developing a self-supervised learning algorithm better aligned with InstructMol than existing methodologies. Could the authors briefly elaborate on the extra compute incurred by the instructor model?

---

> ### Author Rebuttal · Authors · 2024-07-31
>
> Dear Reviewer **nboj**,
>
> Thank you for your thoughtful review and detailed feedback on our paper. We appreciate your recognition of the significance of addressing label scarcity in molecular property prediction and the contributions of our proposed method. We are glad that you found our work timely, important, and well-motivated. Your positive remarks about the clarity of the presentation and the comprehensiveness of our experiments are encouraging. Below, we respond to the key points you raised:
>
> (1) We acknowledge that while our method shows improvements, the performance gains are sometimes accompanied by large standard deviations. **This variability can arise from the inherent complexity of molecular data and the challenges associated with out-of-distribution (OOD) generalization.**
>
> (2) We apologize for any confusion caused by numerical issues and thanks for your detailed question. Here, we report the average increase in ROC-AUC of six classification tasks for **three backbone architectures**, *i.e.*, GIN, GCN, and GAT, instead of the specific improvements for these six tasks. Therefore, there are only three numbers in Lin 201-202. We will revise this section to ensure clarity and provide a complete breakdown of the results for all tasks.
>
> (3) You are correct that **the architecture of the InstructMol model is not necessarily the same as the target molecular model**. InstructMol consists of two components: the main model for molecular property prediction and a separate instructor model that predicts the reliability of pseudo-labels. *The choice of architecture for each component can vary depending on the specific task and dataset.* In our experimental implementation, for simplicity, we directly copy and adopt the same architecture of the instructor as the target model. We will expand the discussion in the Appendix to explain the construction of the InstructMol model as you wish.
>
> (4) Regarding the development of a self-supervised learning algorithm better aligned with InstructMol, we recognize that this is an area ripe for further research. As for the additional computational cost incurred by the instructor model, it includes not only the training of an additional model but also the iterative process of pseudo-label assignment. Therefore, the entire computational expense would be $(1 + 1/k)$ times ($k$ is the update frequency) more than the conventional SSL methods, which rely on a single target model for molecular property prediction. However, *this computational burden can be reduced and the training speed would be significantly accelerated if we select a lightweight instructor architecture (e.g., shallow layers and a smaller hidden dimension with a much smaller model size).* We will include a more detailed discussion of this computational cost, along with potential optimizations or alternatives that could mitigate it in the Limitation part.

---

> > ### Comment · Reviewer_nboj · 2024-08-13
> >
> > I would like to thank the authors for their response. Though the calculation of the average increase in ROC-AUC is still not clear to me (e.g., compared with which baseline model),  hopefully they will clarify further in their revisions. I will keep my score as it is.

---

### Official Review · Reviewer_dbD2 · 2024-07-16

**Soundness:** 3
**Presentation:** 2
**Contribution:** 3
**Rating:** 5
**Confidence:** 4

**Summary:**

The authors develop a method, called InstructMol, for adding pseudo-labels to any training task by including an "instructor" that is trained to discriminate real labels from pseudo-labels, and whose uncertainty is used to modulate the training loss for the primary predictors. The authors show that adding this instructor model improves overall performance over similar methods across a number of property prediction tasks. The authors also show that pretraining delivers state of the art results on the MoleculeNet benchmarks.

**Strengths:**

- novel general pseudo-labeling method
- novel loss for extracting signal from all pseudo-labels, even when the estimated uncertainty is high
- top results amongst comparative models, especially the results in Table 1

**Weaknesses:**

- Using GIN, GAT and GCN for the results in Table 1, but then GEM for those in Table 3, makes it feel like the results are cherry-picked, especially since the GEM+InstructMol results in Table 3 are comparable to and within the error of some of the other methods.
- It is unclear how the results of Figure 4 were obtained. The reader assumes these are comparable to the GEM+InstructMol of Table 3, but this should be better explained.
- The distance of the 9 molecules examined in the real world drug discovery section to the training set should be examined, even if the fact that these were patentable suggests they are dissimilar from known molecules. Also, there are likely many more examples like this that could have been provided more prospective evaluation, and only showing one makes the reader again wonder if the example is cherry-picked.
- It would have been useful to see similar plots to those in Appendix C for other uncertainty estimation methods as a way of showing that InstructMol learns better separation between real and pseudo labels.

**Questions:**

- I can't tell from the paper if the benefit that comes from InstructMol is due simply to training on more data or for more iterations, simply due to having more labeled data. In which of the experiments is this possibility controlled for? Is it Table 1 where other pseudo-label methods are compared to? If such a control exists, will you please make this explicit in the paper? If not, this seems a critical experiment to run.
- Why not use TDC benchmarks instead of MoleculeNet? While the datasets are obviously similar, TDC has done some additional work to clean them, provide reasonable splits, etc.

---

> ### Author Rebuttal · Authors · 2024-08-01
>
> Dear Reviewer **dbD2**,
>
> Thank you for your detailed review and thoughtful comments on our paper. We appreciate your recognition of the strengths, particularly the novelty of our pseudo-labeling method, the innovative loss function for utilizing all pseudo-labels, and the strong performance shown in Table 1. We are pleased that you found our pseudo-labeling approach and loss function novel and effective. Your acknowledgment of the competitive results we achieved, especially in Table 1, is encouraging. Below, we address the concerns and questions you raised:
>
> (1) Your question about the potential benefits from training on more data or iterations is important, and it has also been raised by Reviewer UbNR. First, in response to your query regarding Table 1, we indeed compared InstructMol with other pseudo-labeling methods under controlled conditions to isolate the effects of our approach from simply having more labeled data. For example, UPS [A] introduces an uncertainty-aware pseudo-label selection framework, which enhances pseudo-labeling accuracy by significantly reducing noise during the training process.
>
> Second, by ruling out the benefit of merely having more labeled data, we attribute the success of InstructMol to two key factors. (1) The target model assigns more accurate and reliable pseudo-labels with the assistance of the instructor. (2) The instructor plays a crucial role in evaluating the reliability of these pseudo-labels and influences their contribution to the loss calculation, as detailed in Equation 2.
>
> InstructMol implements a cooperative-yet-competitive learning scheme, which synergistically enhances both the main molecular prediction task and the companion label confidence prediction task. This approach fosters collaboration between the two tasks by exchanging crucial information: the main task contributes prediction loss data, while the companion task provides label confidence measures. This interaction is a significant factor in the observed performance improvements of our method.
>
> **[A]** In Defense of Pseudo-Labeling: An Uncertainty-Aware Pseudo-label Selection Framework for Semi-Supervised Learning
>
>
>
> We will explicitly clarify this in the paper, detailing the control measures we used to ensure a fair comparison. If this aspect has not been fully addressed, we acknowledge the need for additional experiments and will work to include such controls or explicitly state any limitations.
>
>
> (2) We selected MoleculeNet for its extensive range of datasets and well-established benchmarks, making it a widely accepted standard for evaluating molecular property prediction models. Numerous prior studies, including GROVER [A], 3D-Informax [B], GraphMVP [C], MolCLR [D], Uni-Mol [E], GEM [F], and GPT-GNN [G], have utilized MoleculeNet, which facilitates direct comparisons of our results with these methods. Furthermore, we used the same scaffold splitting strategy as GEM and Uni-Mol, which is recognized as a challenging and biologically relevant approach, ensuring a robust evaluation of our model's performance.
>
> Beyond the standard MoleculeNet benchmarks, we also evaluated InstructMol using the **Graph Out-of-Distribution (GOOD) benchmark**, which systematically assesses the generalization capabilities of graph-based models. Additionally, we explored the predictive strength of InstructMol on bioactivity using the CHEMBL214_Ki dataset from the **ACE benchmark** [H], providing further insights into its practical applicability in real-world drug discovery scenarios.
>
> In summary, our evaluation strategy aims to be both comprehensive and thorough, covering a wide range of datasets and challenging scenarios. However, we acknowledge the value of the Therapeutic Data Commons (TDC) benchmarks, particularly their enhanced data cleaning and standardized splits. We appreciate your suggestion and will consider incorporating TDC benchmarks in future research to further validate and extend our findings. Thank you for your constructive feedback.
>
> **[A]** Self-supervised graph transformer on large-scale molecular data. NeurIPS 2020.
>
> **[B]** 3d infomax improves gnns for molecular property prediction. ICML 2022.
>
> **[C]** Pre-training molecular graph representation with 3d geometry. ICLR 2022.
>
> **[D]** Molecular contrastive learning of representations via graph neural networks. Nature 2022.
>
> **[E]** Uni-mol: A universal 3d molecular representation learning framework. NeurIPS 2022.
>
> **[F]** Geometry-enhanced molecular representation learning for property prediction. Nature 2022.
>
> **[G]** Strategies for pre-training graph neural networks. ICLR 2020.
>
> **[H]** Exposing the limitations of molecular machine learning with activity cliffs. JCIM 2022.

---

> > ### Comment · Reviewer_dbD2 · 2024-08-13
> >
> > Thank you for the rebuttal and attempting to address my questions.
> >
> > Re: question 1 - are you saying that UPS is trained on the same amount of data, or for the same number of iterations, as InstructMol, and so this comparison is the control for that?
> >
> > Why were none of the weaknesses I pointed out addressed?
> >
> > I am currently keeping my score as is.

---

### Official Review · Reviewer_73Xa · 2024-07-29

**Soundness:** 3
**Presentation:** 3
**Contribution:** 2
**Rating:** 6
**Confidence:** 3

**Summary:**

The authors present "InstructMol" which does not require transferring knowledge between multiple domains, which avoids the potential gap between the pretraining and fine-tuning stages. and demonstrate it on real-world molecular datasets and out-of-distribution (OOD) benchmarks.

**Strengths:**

Instructive Learning Framework helps the model the generalize better for out-of-distribution molecular property prediction task.

**Weaknesses:**

The paper mostly focus on GNN as backbone but would be worth discuss more about transformer based model trained on SMILES representation.

**Questions:**

How would Instructive Learning Framework works on  transformer based model trained on SMILES representation?

**Limitations:**

The paper mostly focus on GNN as backbone but would be worth discuss more about transformer based model trained on SMILES representation.

---

> ### Author Rebuttal · Authors · 2024-07-31
>
> Dear Reviewer **73Xa**,
>
> Thank you for your detailed feedback on our InstructMol. We appreciate your insights and suggestions, which are invaluable for refining our work. We are pleased to hear that you found our Instructive Learning Framework effective for improving generalization in OOD molecular property prediction tasks. This was a primary goal of our work, and we're glad it was recognized.
>
> You noted that our paper primarily focuses on GNNs and suggested that it would be beneficial to discuss the application of our framework with transformer-based models trained on SMILES representations. This is an excellent point. While we concentrated on GNNs due to their strong performance in molecular graph representations, we acknowledge that transformers have shown promising results in modeling sequential data like SMILES.
>
> To address this, we are currently exploring how our InstructMol can be adapted for transformer-based architectures. To be specific, we leverage an open source Transformer-based algorithm -- SMILES Transformer (ST) [A] as the backbone and evaluate the impact of our instructive learning framework on 10 datasets from MoleculeNet, which is also adopted in its original paper but *with a different splitting strategy*. Note that we adopt a 250K unlabeled dataset for SSL. Preliminary results (see Table below) suggest that the framework's principles are generalizable and can indeed enhance the performance of transformer models on SMILES data. We plan to include a discussion of these findings in the final version of the paper, detailing how the framework's principles can be adapted and optimized for different model architectures.
>
> | Dataset | ESOL $\downarrow$ | FrSlv $\downarrow$ | Lipo $\downarrow$ | MUV $\uparrow$ | HIV $\uparrow$ | BACE $\uparrow$ | BBBP $\uparrow$ | Tox $21 \uparrow$ | Sider $\uparrow$ | ClinTox $\uparrow$ |
> | :---: | :---: | :---: | :---: | :---: | :---: | :---: | :---: | :---: | :---: | :---: |
> | ST | 1.144 | 2.246 | 1.169 | 0.009 | 0.683 | 0.719 | 0. 900 | 0.706 | 0.559 | 0.963 |
> | ST + InstructMol | **1.013** | **2.089** | **1.058**|  **0.025**|  **0.704**|  **0.733**|  **0.908**|  **0.742**|  **0.570**|  **0.971**|
> ---------------------------------------------
> [A] SMILES Transformer: Pre-trained Molecular Fingerprint for Low Data Drug Discovery. 2019

---

> > ### Comment · Reviewer_73Xa · 2024-08-13
> >
> > Thanks the author for checking the results on SMILES Transformer! I am pleased to see that it helps not only on GNN. I am keeping my score as is.

---

> > > ### Author Response · Authors · 2024-08-13
> > > **Thanks for Feedback**
> > >
> > > Dear Reviewer 73Xa,
> > >
> > > Thank you for your positive feedback and your continued recommendation for acceptance. We are delighted to hear that our additional experiments have addressed your concerns. Thank you once again for your valuable time and insights!
> > >
> > > Best regards,
> > >
> > > Authors

---

### Decision · Program_Chairs · 2024-09-25

**Decision:**

Accept (poster)

**Comment:**

The paper presents an instructive learning framework that uses large-scale unlabeled data to enhance molecular property prediction. Unlike traditional methods requiring knowledge transfer between domains, InstructMol operates within a single domain, avoiding discrepancies between pretraining and fine-tuning stages. The framework incorporates a pseudo-labeling mechanism supported by an instructor model that estimates the reliability of these labels. The authors demonstrate InstructMol on molecular datasets and OOD benchmarks.

The reviewers provided a balanced evaluation of the paper, highlighting several strengths and weaknesses:

**Positive Points:**
* The InstructMol framework is praised for its approach to leveraging unlabeled data for molecular property prediction, particularly its ability to handle OOD scenarios effectively.
* The method addresses a critical problem in biochemical research, namely data scarcity, and demonstrates its applicability through extensive benchmarking, achieving competitive results.

**Negative Points:**
* Several reviewers noted the need for a more detailed discussion on the computational complexity and resource requirements of the InstructMol framework, particularly its scalability to large datasets.
* Concerns were raised about potential overfitting issues due to the iterative use of pseudo-labels and method's robustness to noise introduced by additional training data.
* There was some critique regarding the clarity and organization of specific sections, such as the related work, and the need for more detailed analysis of certain experimental results and their implications.

After considering the reviewers' feedback and the authors' rebuttals, I recommend accepting the paper at NeurIPS. While the reviewers identified areas for improvement, the authors' rebuttals were thorough and addressed many of the concerns. The weaknesses, such as computational complexity and potential overfitting, are acknowledged but do not detract significantly from the overall impact and novelty of the work.